# Isolation of subpollen particles (SPP) of birch: SPP are potential carriers of ice nucleating macromolecules

Julia Burkart[1], Jürgen Gratzl[1], Teresa M. Seifried[2], Paul Bieber[2], Hinrich Grothe[2]

[1] Faculty of Physics, University of Vienna, 1090 Wien, Austria
[2] Institute of Materials Chemistry, TU Wien, 1060 Wien, Austria

*Correspondence to*: Julia Burkart (julia.burkart@univie.ac.at)

**Abstract.** Within the last years pollen grains have gained increasing attention due to their cloud forming potential. Especially the discovery that ice nucleating macromolecules (INM) or subpollen particles (SPP) obtained from pollen grains are able to initiate freezing has stirred up interest in pollen. INM or SPP are much smaller and potentially more numerous than pollen grains and could significantly affect cloud formation in the atmosphere. However, INM and SPP are not clearly distinguished. This has motivated the present study, which focuses on birch pollen and investigates the relationship between pollen grains, INM and SPP. According to the usage of the term SPP in the medical fields, we define SPP as the starch granules contained in pollen grains. We show that these insoluble SPP are only obtained when fresh pollen grains are used to generate aqueous extracts from pollen. Due to the limited seasonal availability of fresh pollen grains almost all studies have been conducted with commercial pollen grains. To enable the investigation of the SPP we develop an alternative extraction method to generate large quantities of SPP from commercial pollen grains. We show that INM are not bonded (i.e. can be washed off with water) to SPP. Further, we find that purified SPP are not ice nucleation active: after several times of washing SPP with ultrapure water the ice nucleation activity completely disappears. To our knowledge, this is the first study to investigate the ice nucleation activity of isolated SPP. To study the chemical nature of the INM, we use fluorescence spectroscopy. Fluorescence excitation-emission maps indicate a strong signal in the protein range (maximum around $\lambda_{ex} = 280$ nm and $\lambda_{em} = 330$ nm) with all ice nucleation active samples. In contrast, with purified SPP the protein signal is lost. We also quantify the protein concentration with the Bradford assay. The protein concentration ranges from 77.4 µg ml$^{-1}$ (Highly concentrated INM) to below 2.5 µg ml$^{-1}$ (purified SPP). Moreover, we investigate the connection between proteins and ice nucleation activity by treating the ice nucleation active samples with subtilisin A and urea to unfold and digest the proteins. After this treatment the ice nucleation activity clearly diminished. The results indicate a linkage between ice nucleation activity and protein concentration. The missing piece of the puzzle could be a glycoprotein, which exhibits carboxylate functionalities, can bind water in tertiary structures and displays degeneration and unfolding of its secondary structure due to heat treatment or reaction with enzymes. Even though purified SPP are not ice nucleation active they could act as carriers of INM and distribute those in the atmosphere.

## 1 Introduction

Aerosol particles in the atmosphere are climatically relevant if they affect cloud microphysics and formation of precipitation or directly scatter solar radiation. The aerosol effect on clouds is still one of the largest uncertainties in our climate system (Boucher, 2013) and the role of bioaerosols including pollen is far from being understood (Fröhlich-Nowoisky et al., 2016). Pollen are a subset of bioaerosols such as bacteria, viruses, algae, fungi and plant debris found abundantly in the atmosphere during the blooming season of plants. Pollen is the male gametophyte that contains vegetative and generative cells essential for fertilization and reproduction of a plant. In particular, wind pollinated plants such as early flowering trees (e.g. birch) and grasses produce pollen in large quantities and rely on wind for dispersal and transportation to the female plant, where fertilization occurs. Pollen are thus distributed in the lower atmosphere and can also travel long distances when meteorological conditions are favourable (Damialis et al., 2017; Sofiev et al., 2013; Skjøth et al., 2007). Typical concentrations of pollen

grains of a specific type range from 10 m$^{-3}$ to a few hundred or even thousands per m$^3$ on a daily average. Peak concentrations in the vicinity of plants can assume a few 100 000 m$^{-3}$ (Jochner et al., 2015; Šikoparija et al., 2018). Forests of the Northern Hemisphere are estimated to be large contributors to atmospheric pollen concentrations e.g. birch pollen concentrations outside the canopy of birch forests are in the range of 10 000 m$^{-3}$ during spring (Williams and Després, 2017). The ability of pollen to induce liquid droplet and ice formation has been documented in several studies already years ago. Entire pollen grains were shown to act as giant cloud condensation nuclei (CCN) (Pope, 2010) and also as ice nucleating particles (INP) (Diehl et al., 2002; Diehl et al., 2001; von Blohn et al., 2005). A summary of the role of pollen and other biological particles in cloud physics is given by Möhler and colleagues (Möhler et al., 2007). However, due to their large sizes (10-100 μm), leading to short residence times in the atmosphere, their impact on cloud microphysics was estimated to be negligible on a global scale (Hoose et al., 2010a,b).

Only discoveries of recent years have demonstrated that not only the entire pollen grain has the ability to initiate ice formation, but the presence of macromolecules originating from pollen grains is sufficient to induce heterogeneous ice nucleation (Pummer et al., 2012; Pummer et al., 2015). In their study, Pummer and co-workers (Pummer et al., 2012) suspended entire birch pollen grains in water for several hours. The solution was then decanted and filtrated yielding what is called pollen washing water. The washing water is shown to induce ice formation at similar temperatures as the entire pollen grains. The ice activity of pollen grains is thus attributed to the existence of ice nucleating macromolecules (INM) that can be separated from pollen grains (Pummer et al., 2015). Such INM can also be obtained from other parts of a plant e.g. branches, leaves, and barks of a birch (Felgitsch et al., 2018). Aqueous extracts of pollen grains obtained in a similar manner were used in several other studies to investigate the cloud forming potential of many different types of pollen grains (Gute and Abbatt, 2020; Gute et al., 2020; Mikhailov et al., 2019; O′Sullivan et al., 2015; Steiner et al., 2015). In these studies, such aqueous extracts are often referred to as subpollen particles (SPP). The term SPP originates in the medical sciences and is commonly used to describe starch granules with allergenic potential that are contained in pollen grains (Bacsi et al., 2006; Schäppi et al., 1999; Sénéchal et al., 2015). The term starch granule refers to the main component starch, which is a polysaccharide and functions as an energy storage unit in plant cells (Baker and Baker, 1979; Buléon et al., 1998; Hancock and Tarbet, 2000). In studies related to atmospheric sciences, the term is used in a less defined and sometimes confusing way. Aqueous extracts of pollen grains are simply referred to as SPP even though they might not contain any particulate material or particles created artificially by atomizing solutions of aqueous pollen extracts are denoted SPP.

A recent model simulation estimates that the presence of SPP in the atmosphere could suppress precipitation in clean continental clouds by about 30% (Wozniak et al., 2018). However, estimations of the amount of such material in the atmosphere is based on rough assumptions, since data is missing. One reason for the lack of data is that little is known about the nature and composition of INM and, as a consequence, direct measurement methods that can identify INM in the atmosphere are not available. Up to date only a few studies investigated the chemical nature of INM and found that INM contained in birch pollen washing water are water soluble and are likely composed of polysaccharides (Dreischmeier et al., 2017; Pummer et al., 2012; Pummer et al., 2015). However, there is also evidence that the ice nucleating activity stems from proteinaceous substances, e.g. Tong and colleagues show that the ice activity of birch washing water diminishes if proteinaceous components are extracted (Tong et al., 2015).

One major difficulty in identifying INM is related to the complexity of a pollen grain. In fact, a pollen grain is a highly complex particle composed of different parts and materials (Fægri and Iversen, 1992). Figure 1 illustrates a birch pollen grain and its main parts. The actual cell is protected by a robust outer shell made of two layers: the exine (outer layer) and the intine (inner layer). The exine is composed of the mechanically resistant and chemically inert biopolymer sporopollenin, while the intine, composed of cellulose and pectin, is a more fragile membrane. The cell is filled with cytoplasmic material including proteins,

lipids and polysaccharides such as starch. Many pollen types, such as birch pollen grains, also contain pores. Pores are spots where the exine is missing and thus give access to the cell membrane (intine). Under humid conditions most fresh pollen grains are likely to expel cytoplasmic material, a process often referred to as pollen rupture. In literature two mechanisms are documented by which pollen grains can release cytoplasmic content including starch granules from their interior. First, entire pollen grains can rupture by osmotic shock during hydration (D'Amato et al., 2007; Suphioglu et al., 1992; Taylor and Jonsson, 2004). Second, pollen grains can be stimulated to germinate and grow pollen tubes (Grote et al., 2003; Schäppi et al., 1997). The pollen tubes rupture at their tips just as they would during fertilization on a female stigma and expel genetic and cytoplasmic material. This process is referred to as abortive germination (Grote et al., 2003). In a more recent laboratory study Sénéchal and co-workers showed that SPP are also released from birch and cypress pollen after the grains were exposed to humidity or to $NO_2$ (Sénéchal et al., 2015). Mechanical rupture caused by wind induced impaction might also generate SPP (Visez et al., 2015).

These processes have been examined since in several studies from the medical fields e.g. (Grote et al., 2003; Schäppi et al., 1997; Staff et al., 1999; Suphioglu et al., 1992; Taylor et al., 2004). Pollen rupture is hypothesized to occur in the atmosphere and offers an explanation of the presence of allergens in the fine aerosol fraction ($< 5\mu m$) i.e. the presence of allergens detached from pollen grains. For example, birch pollen grains were shown to germinate on leaves after light rain and release starch granules (Schäppi et al. 1999). Simultaneously a rise in the allergen concentration in the air was measured without the presence of pollen grains (Schäppi et al., 1999; Schäppi et al., 1997). SPP are also hypothesized to cause the phenomenon of thunderstorm asthma (Taylor et al., 2007; Taylor and Jonsson, 2004). Thunderstorm asthma is the observed coincidence of severe asthma epidemics and thunderstorms (e.g. (Thien, 2018)). Due to their large size, pollen grains are efficiently trapped in the upper airways when inhaled. In contrast, SPP can penetrate deep into the lung and trigger asthma. Cases of thunderstorm asthma have been documented worldwide (D'Amato et al., 2019). An increase of submicron particles of biological origin possibly connected to pollen grains during rain events or thunderstorms has also been observed in a few recent studies not related to allergens that use chemical tracers and measurements of fluorescence particles as an indication for biological particles possibly originating from pollen grains (Hughes et al., 2020; Rathnayake et al., 2017; Huffman et al., 2013).

In this study, we investigate SPP of birch pollen grains. The aim of our study is to shed light on the differences between INM and SPP. As discussed above, in atmospheric literature the term SPP is usually used in a rather unspecific sense to simply describe the remaining materials when aqueous extracts of pollen grains are generated. For the purposes of our study, we follow the usage of SPP as done in medical literature. Purified SPP are composed of starch and are contained in the pollen grains as energy storage units. Starch is a polysaccharide made of amylose and amylopectin (Hancock and Bryon 2000, Buléon et al., 1998) inside the amyloplast membrane (Matsushima et al., 2016). Accordingly, we define SPP as the starch granules contained in the cytoplasm. With the extraction methods commonly used and predominantly applied to commercial pollen grains, SPP are not obtained. We emphasize that commercial pollen grains do not rupture and do not expel SPP. We therefore develop a method to extract SPP and separate those from other cytoplasmic materials. We thereby aim to answer the question whether SPP are ice nucleation active and if INM are linked to SPP. To our knowledge this is the first study to investigate the ice nucleating activity of SPP thoroughly extracted from pollen grains. Additionally, we use fluorescence spectroscopy to gain

chemical information about the ice active substances and the Bradford assay to quantify protein concentrations. To clarify the role of proteins in heterogeneous freezing we conducted a specific enzymatic digestion/ protein unfolding experiment.

## 2 Experimental Section

### 2.1 Pollen samples

We used commercially purchased birch pollen samples. The samples were purchased from ThermoFisher Scientific Allergon (www.allergon.com) and specified by the company as pollen from *Betula pendula* (common name: silver birch). Pollen are collected from trees in Southern Sweden and are carefully purified from other plant materials after sampling. The pollen sample is reported to contain less than 2% of plant parts other than pollen grains. At the time of purchase the birch pollen grains were already 1 year old. Freshly harvested pollen samples were collected from a birch tree at the Donaukanal (location birch tree: 48.237480, 16.362990) in Vienna. Collection took place on April 9, 2021, when birch trees were reported to be ready to bloom by the Austrian pollen forecast service (www.pollenwarndienst.at). Additionally, phenological observations at the sampling site~~location~~ were performed (i.e. check for clearly visible anthers on the catkins) to confirm the maturity of the pollen grains.

### 2.2 Extraction of SPP

In order to extract SPP of birch pollen grains we developed an extraction process that starts with cracking the pollen grains so that the cytoplasmic content is available even without the process of pollen rupture. Fresh birch pollen grains naturally rupture when soaked with water and release their cytoplasmic content including starch granules. With commercially purchased pollen this ability is almost lost due to aging and desiccation of pollen and the loss of viability. Therefore, we use a mixer mill (MM400, Retsch GmbH, Haan, GER) to crush the grains. The extraction process is illustrated in Figure 2. Crushing the pollen grains with the mixer mill is the first step of the extraction method (Figure 2a, step 1). In order to crush the grains with the mixer mill we blend 0.5 g pollen grains with 2 ml ultrapure water (MilliQ, 18.2 MΩ) and pour the water pollen suspension into the grinding jar including one single ball as a grinder. The mixer mill is operated for 1 minute at 25 Hz (step 1). As a result, 30-40 % of the pollen grains are cracked and cytoplasmic material and SPP are gained (see Figure 2b). It should be noted that the pollen wall cracks but does not fragment into very small pieces (below ~ 10 µm). We found very few wall fragments after this filtration process (analysed in electron microscopic pictures). Apart from the SPP, the cytoplasmic content appears soluble forming amorphous structures when dried.

In a second step, we use filter paper with 10 µm pore size to filter pollen wall fragments and remaining undamaged pollen grains. Particulate material larger than 10 µm is retained by the filter, while all other materials pass through. The resulting extraction product is referred to as sample A. Sample A contains SPP, other cytoplasmic materials (mostly soluble) and few small wall fragments.

In step 3, sample A is filtrated with a syringe filter (0.2 µm pore size, Nylon membrane, sterile) i.e. particles larger than 0.2 µm are retained by the filter. Sample B refers to the solution without SPP.. In step 4, the filter with the remaining material is rinsed with ultrapure water until most of the soluble material is washed off. The rinsing is done stepwise with 1 to 70 ml ultrapure water. After passing through the filter the filtrate is called sample C. The ice nucleation activity of each sample C fraction was tested for each rinsing step (note that the rinsing water was not pooled). As the purpose of this extraction was to test if INM can be separated from SPP, rinsing was continued as long as sample C was ice nucleation active. Rinsing was stopped after 70 ml when the ice nucleation activity of sample C had disappeared.

After rinsing, the filter is reversed and flushed with ultrapure water (step 5). A suspension of SPP in ultrapure water is obtained. This sample is referred to as sample D (Figure 2c) and used to test the ice nucleation activity of SPP.

## 2.3 Determination of the geometric size distribution

The geometric size of the SPP was determined from images taken with a field emission gun scanning electron microscope (FEG-SEM; Zeiss Supra 55 VP). The shape of the SPP was approximated by a cylinder capped with hemispheres at each side with the diameters of the hemispheres being equal to the width of the cylinder (Figure 3). The width s and length l of the cylinder was measured manually with the help of an image analysis software (SmartTIFF V1.0.1.2) and the volume was calculated. This was done for 326 particles in total. The volume was then used to determine the volume equivalent diameter of the SPP i.e. the diameter of a sphere having roughly the same volume as the irregular shaped SPP. Based on the approximated shape the volume equivalent diameter $d_v$ was calculated:

$$d_v = s \sqrt[3]{\frac{1}{2}(\frac{3l}{s} - 1)} \qquad (1)$$

Additionally, the aspect ratio was used to describe the sphericity of the particle. The aspect ratio is defined as width/length and is 1 for spherical particles and smaller than 1 for elongated particles.

## 2.4 Ice nucleation measurements

INM content from birch pollen was quantified in immersion freezing mode by using the Vienna Optical Droplets Crystallization Analyzer (VODCA) setup (a detailed description of the experiment is given in Felgitsch et al. 2018, Zolles et al., 2015 and; and Pummer et al., 2012): An emulsion of 2 µl sample solution and 4 µl inert oil-mixture (10 wt% lanolin, anhydrous, VWR Int., Radnor, PA, USA; 90 wt% paraffin, light, pure grade, AppliChem GmbH, Darmstadt, GER) is prepared by mixing with a pipette tip on a microscopic glass slide and transferred into a cryo-cell. Sample emulsions are cooled with a rate of 10 °C min$^{-1}$. The freezing process was monitored by videos at four different spots of each sample glass slide via a microscope camera (MDC320, Hengtech, GER). On each spot about 20 droplets in the corresponding size range are observed. The fraction of frozen droplets ($n_{frozen}/n_{total}$) were plotted as a function of the temperature. Furthermore, the cumulative nucleus concentration (CNC) was determined using equation 2 (Vali, 1971):

$$CNC(T_i) = -\frac{ln\left(1 - \frac{n_{frozen}}{n_{total}}\right)}{V} \cdot d \qquad (2)$$

where $n_{frozen}$ is the number of frozen droplets at the given temperature, $n_{total}$ the total number of droplets, V the droplet volume and d the dilution factor. CNC ($T_i$) indicates the number of INM per unit volume actively present above the temperature $T_i$. Note that one ice nucleus can also be an aggregate of more than a single molecule (Qui et al., 2019). To compare the values of INM from different samples we chose $T_i$ at -25°C since most biogenic INM are active above that temperature (Kanji et al., 2017; Murray et al., 2012; Pummer et al., 2015) and -34°C since homogeneous ice nucleation starts at -35°C with the VODCA setup. Thus, the CNC (-34°C) value includes all heterogeneous freezing events. High concentrated samples were diluted in the experiment to prevent underestimation of INM contents and the dilution factor (d) was included in equation 2 (see Vali, 1971). Only droplets with a diameter between 15 and 40 µm (droplet volume: 1.8-34 pl) are counted in the evaluation and the average volume of 8.2 pl was used to calculate CNC values. The selected size range is relevant for atmospheric cloud droplets. The temperature uncertainty of VODCA is 0.5 °C (Zolles et al, 2015). Furthermore, we calculated the counting error and performed a Gauß error propagation as described in Kunert et al. (2018). Accordingly, data points with errors larger than 100% were excluded in the graphs .

## 2.5 Fluorescence spectroscopy

Autofluorescence active materials such as SPP and respective extracts (see Figure 2) were characterized by fluorescence spectroscopy using a FSP920 spectrometer (Edinburgh Instruments, UK), equipped with a Xe900 xenon arc lamp (450 W) and a S900 single photon photomultiplier. Sample solutions were measured in a quartz glass cuvette (500 µL, Hellma Quartz (Suprasil®), GER) using a designated cuvette optic. The excitation light beam is arranged in a 90° angle relative to the detector. The software F900 allowed recording excitation-emission maps (excitation from 220 to 400 nm, emission from 320 to 500 nm). The excited state was held for 0.25 s dwell time and step width of the monochromators was set to 2 nm. To avoid first and second order excitation (Figure 7, grey area), we used a 295 nm low-pass filter (Stablife Technology®, Newport, USA) and an offset of 10 nm. Two samples (sample A and B, see Figure 2) were highly fluorescent active and thus, diluted during the measurements, to minimize the influence of quenching agents. Obtained excitation-emission maps were normalized to 5.0 x $10^4$ counts.

## 2.6 Quantitative protein analysis

The protein concentration of the sample solutions was determined by the common protein assay firstly described by Marion M. Bradford (Bradford, 1976). The method is based on a protein-dye binding reaction. The dye-reagent Coomassie Brilliant Blue forms a complex with proteins in solutions and shifts the absorption maximum from 465 to 595 nm. A standard curve ranging from 2.5 to 25 µg ml$^{-1}$ was prepared by diluting an albumin standard (Thermo Scientific, GER) with ultrapure water. The samples from step A and B (see section section 2.2) were diluted 1:5 since their concentration was out of test specifications. We pipetted 150 µl of each sample, standard and blank (ultrapure water) directly into a microtiter microplate (MaxiSorp, Nunc-Immuno, Thermo Scientific, GER). By using a stepper pipette, 150 µl of Coomassie reagent (Thermo Scientific, GER) were added to each cavity. The plate was shaken with a microplate shaker (PMS-1000i, Grant Instruments, UK). After incubating for 10 minutes at room temperature, we measured the absorbance at 600 nm with a photometer (Sunrise, Tecan, CH). For evaluation, the average blank value was subtracted from the absorption values. For the standard curve a linear regression was determined and the total protein content of the sample was calculated.

## 2.7 Enzymatic and chemical treatment

Sample B was treated with (i) subtilisin A, (ii) urea and (iii) a mixture of subtilisin A and urea. The enzyme subtilisin A is a protease, which belongs to the Serine S8 Endoproteinase family (Hedstrom, 2002). It denatures proteins with a broad specificity by hydrolyzing peptide bonds (Rawlings et al., 2010). Highly concentrated urea, in contrast, is a chaotropic reagent which does not cut covalent bonds but unfolds proteins by weakening the hydrogen bond system.

First, a 100 mM Tris buffer (Sigma Aldrich, 252859, St. Louis, MO, USA) with a pH of 8.4 (adjusted with a 2 M HCl) was prepared. Subtilisin A was dissolved in the Tris buffer with a resulting concentration of 2 mg ml$^{-1}$. Further, we prepared a 8 mol/l urea solution (Sigma Aldrich, 33247, St. Louis, MO, USA) using the buffer as a solvent. For the incubation, we prepared four samples: 50 µl of sample B were mixed with (1) 210 µl Tris buffer (serving as a control sample, to monitor whether temperature changes the INA), (2) 10 µl subtilisin A and 200 µl Tris, (3) 10 µl Tris and 200 µl urea and (4) 10 µl subtilisin A and 200 µl urea. All samples had a total volume of 260 µl resulting in a Tris buffer concentration of 81 mmol/l and 6.2 mol/l urea. The samples were incubated at 37 °C (Heiz-Thermoshaker, Thermo Scientific, Waltham, WA, USA); after 1 h and 6 h, 2 µl sample aliquots were taken and diluted 1:10 with ultra pure water prior to ice nucleation measurements to decrease the freezing point depression of the buffer and urea. Chemicals used in the treatment did not show heterogeneous ice nucleation activity.

## 3 Results and Discussion

### 3.1 Extraction process and size distribution of SPP

The extraction process in this study differs from the usual approach in other studies (e.g. (Gute and Abbatt, 2020; Pummer et al., 2012; Steiner et al., 2015)) especially in one aspect: commercial or stored pollen grains do not germinate nor rupture, and therefore do not release insoluble SPP (starch granules) contained in the cytoplasm. In contrast, we observe that fresh pollen grains directly shed from catkins germinate and rupture when immersed in water or exposed to high relative humidity (>95 %) for several hours.. This process has also been documented in literature (Grote et al. 2003). To ensure that insoluble SPP and material from inside the pollen grains is obtained we first crack the exine of the pollen grains. This is done with a mixer mill. As seen in Figure 2b the exine cracks and gives access to the pollen grain's interior including the starch granules. This step was necessary to mimic the natural behaviour of fresh pollen grains. Figure 4 shows fresh pollen grains that had been exposed to high relative humidity (>95%) for 8 hours. The behaviour of fresh pollen grains immersed in water is also illustrated in the linked video (https://ucloud.univie.ac.at/index.php/s/FuF5SVBfqayb0ta). In both cases particulate, insoluble material can be clearly observed. These insoluble SPP are likely to spread in the atmosphere due to pollen rupture or abortive germination (Schäppi et al. 1999, Grote et al. 2003; Taylor et al 2004, Hughes et al. 2020). With fresh birch pollen we find that both processes take place: pollen rupture (video) and abortive germination (Figure 4). Note that most of the insoluble SPP in Figure 4 are coated with amorphous material that is expelled by the pollen grain at the same time as the SPP. Only a few SPP seem "pure" (the interpretation, however, is limited by the resolution of the electron microscope). Most of the amorphous material probably originates from the cytoplasm, but we cannot exclude that some of the material is also washed off the exine i.e. the pollen grain surface. We find that the ability to rupture is almost entirely lost when freshly harvested birch pollen grains were stored in the lab for a few days to weeks. The highest germination activity (i.e. most pollen grains germinated) was observed when fresh pollen grains were exposed to water on the very same day they were harvested. With commercially purchased pollen grains we did not find any germination activity and also no SPP. In addition, we also treated pollen grains mixed with water up to 1 hour in the ultrasonic bath to see if pollen rupture could be induced this way. However, even after 1-hour ultrasonic treatment we did not find any ruptured pollen grains nor SPP (Figure S1). We emphasize that the usually applied extraction methods, where pollen grains are only left in water and are then filtrated, do not actually yield SPP unless very fresh pollen grains are used. The usual extraction method likely yields only the most soluble components. For example, it is known that some highly soluble proteins (mostly allergenic ones) migrate to the pollen surface within seconds to minutes upon hydration even without pollen rupture (e.g. Vrtala et al. 1993). The exine of birch pollen contains microchannels that enable such an exchange (Diethart et al. 2007). Another study also documents the passage of proteins contained in the cytoplasm through the intact cell membrane (Hoidn et al. 2005). While insect pollinated plants produce pollen with a thick lipid containing pollenkitt (coating of the excine) that functions as a barrier to water soluble components, this pollenkitt is almost entirely absent with pollen from wind pollinated plants such as birch pollen (Diethart et al. 2007). Our extraction method is unique as it guarantees access to less soluble substances and might be closer approximation to the processes in the atmosphere than the usual applied extraction method. Our method offers as well the possibility to study the ice nucleation activity of isolated SPP for the first time and to investigate whether INM are connected to SPP.

We also analysed the geometric size of the SPP gained by our extraction method and calculated volume equivalent diameters (Figure 3, equation (1)). The volume equivalent diameters of SPP range from 0.2 to 2.5 µm and are roughly normally distributed (Figure 5a). The mean value of the distribution as well as the maximum of the gauss-fit is about 1.1 µm. To describe

the sphericity of the SPP the aspect ratio was calculated. It ranges from 0.27 to 1 where 1 describes a perfect sphere. The aspect ratio (Figure 5b) is roughly log-normal distributed with a maximum at about 0.5 confirming that most SPP have a rather elongated form as can be seen in Figure 2c. In fact, the length of the largest SPP reaches up to 4 µm. Considering a density of 1.6 g m$^{-3}$ for SPP (Dengate et al., 1978) and neglecting the non-spherical shape, the aerodynamic diameter would be a factor 1.26 larger than the volume equivalent diameter. The aerodynamic diameter is essential to describe the behaviour of particles in the atmosphere. With this rough estimation we can conclude that the aerodynamic diameter of the SPP ranges from 0.25 to 3.2 µm. It should be noted that we used a filter with 0.2 µm pore size and that smaller particles might have been lost in the extraction process.

## 3.2 Ice nucleation activity

We analysed the ice nucleation activity of all samples obtained during the SPP extraction process (sample A, B, C and D, see Figure 2). The goals of these measurements were: (1) to gain information about the location of the INM within the pollen grain and (2) to investigate whether SPP are ice nucleation active.

In step 4, the syringe filter was rinsed with up to 70 ml of ultrapure water. Rinsing was continued until the ice nucleation activity of the rinsed material (sample C) was completely lost i.e. only homogeneous freezing occurred at temperatures below -35°C. Here, the rinsing steps are labelled C01 to C70. The number indicates the water volume in ml used to rinse the sample up to this step i.e. C01 and C70 mean that the sample had been rinsed in total with 1 ml and 70 ml of ultrapure water, respectively. We analysed the ice nucleation activity of all C01 to C70 samples obtained during the washing process. Freezing curves of all samples are shown in the SI (Figure S2). Figure 6 shows selected samples only to avoid an overcrowded figure. The freezing curves follow the typical pattern (inclining curves and formation of a plateau) that results from diluting an ice active substance (Felgitsch et al., 2018). Sample A, B as well as C01 clearly exhibit heterogeneous ice nucleation. The concentration of INM active above -34°C was 13.2 pl$^{-1}$ for sample A, 13.1 pl$^{-1}$ for sample B and 8.7 pl$^{-1}$ for sample C01 (Figure 6 a). After sample C10 the ice nucleation activity rapidly diminishes but only after 70 ml of washing homogeneous freezing was the dominant process in the experiment (99 % of the observed droplets froze homogeneously). After the last washing step, purified SPP (sample D) were obtained by reversing the syringe filter and flushing the filter with ultrapure water. Sample D did not show heterogeneous ice nucleation activity. As losses of SPP in the syringe filter are expected during the washing process we also conducted a control experiment, where we generated a sample with highly concentrated SPP. In the control experiment we stopped the washing process after 30 ml and further separated SPP from solutes via centrifugation (1000 rcf, 3 min). A picture of the concentrated SPP precipitation is shown in the SI (Figure S3). After several steps of centrifugation and cleaning the precipitate with ultrapure water, even the accumulated SPP did not exhibit ice nucleation activity (see SI, Figure S4). The result indicates that purified SPP are not ice nucleation active. The results thereby show that INM can be separated from SPP and are not tightly attached to the SPP. However, based on our experiments we cannot make a definite statement on the type of bonding.

## 3.3 Chemical Analytics

In order to gain information about the chemical nature of the INM we analysed the extracted samples using fluorescence spectroscopy. Since we found that the starch contained in pollen grains is not ice active, we focused this analysis on the emission/excitation wavelength range where proteins are expected. Biological ice nucleation is often linked to the presence of proteins, acting as INP (Maki et al., 1974; Wilson et al., 2006). The amino acids tryptophan, tyrosine and phenylalanine, which are present in natural proteins, contain excitable π-electrons. Thus, proteins show auto-fluorescence when excited at around 280 nm with a Stokes shift of about 50 nm (Pöhlker et al., 2012). In Figure 7 a-c fluorescence excitation-emission maps of our samples show high intensity at $\lambda_{ex}$ 280 nm and $\lambda_{em}$ 330 nm. This maximum clearly indicates the presence of proteins. The

fluorescence intensity at the maximum decreases with decreasing CNC (Table 1). Furthermore, we did not see any signal in this range for purified SPP (Figure 7 d), leading us to the conclusion that no proteins are detectable in that final fraction anymore. The thin line still visible in Figure 7 d is the Raman signal of water. We conclude that the ice nucleation of our samples might be linked to proteins, whereas the SPP mainly composed of starch are not ice nucleation active.

Even though proteins can show strong fluorescence signals, light absorbing substances in the extract might lead to quenching effects (Papadopoulou et al., 2005) that decrease the protein signal. More specific measurements of protein content can be carried out using UV-Vis spectroscopy after staining the proteins with Coomassie Brilliant Blue (Bradford assay (Bradford, 1976)). Quantification via Bradford assay gave protein concentration of 77.4 µg ml$^{-1}$ for sample A, 25.8 µg ml$^{-1}$ for sample B and 5.3 µg ml$^{-1}$ for C01. Values for D were lower than the limit of quantification (2.5 µg ml$^{-1}$). A summary of the determined values for samples analysed is given in Table 1. Comparing the protein concentration to CNC(-25°C), representing the biological region (Kanji et al., 2017; Murray et al., 2012; Pummer et al., 2015), shows a general trend: a higher protein concentration coincides with higher CNC(-25°C) values. For lower temperatures (i.e. CNC (-34°C)) the dependency on the protein concentration is less pronounced since non-proteinaceous materials become ice nucleation active in the temperature window between -25°C and -34°C.

To further test the hypothesis whether proteins play a role in the ice nucleation activity we conducted an enzyme digestion /protein unfolding experiment using subtilisin A and urea. Treating ice nucleation active samples with enzymes (e.g. Kozloff et al., 1991, Pummer et al., 2012, Felgitsch et al., 2019), chaotropic reagents (e.g. Pummer et al., 2012, Fröhlich-Nowoisky et al., 2015) or strong oxidizer (e.g. H$_2$O$_2$, Gute et al., 2020) to investigate the nature of ice nuclei has been performed in the past. The experiment conducted in this paper is inspired by a publication from Felgitsch et al., 2019 where they investigated the role of proteins in ice nucleation active extracts from perennial plants.

Results show that after incubating sample B for 1 h at 37 °C, the ice nucleation activity of samples containing urea decreased. For urea alone, 65 % of droplets remained to freeze heterogeneously (see Figure 8 a). However, treating the sample with the combination of subtilisin A and urea lead to an even stronger decrease in INA due to the unfolding activity of urea (50 % of droplets froze heterogeneously). Sample B in Tris (control sample) and the digestion with subtilisin A alone did not show a significant decrease of heterogeneous freezing activity after 1 h treatment. However, after 6 h incubation time, subtilisin A seems to slightly influence the INA and the freezing curve is dropping at around -22 °C (see Figure 8 b). Nevertheless, the urea treatment decreased the ice nucleation ability even more. Further, the strongest influence of INA is clearly derived from the mixture of urea and subtilisin A. Again, this can be explained by the unfolding effect of urea in combination with the cleavage of peptide bonds by subtilisin A. In addition, the freezing onset temperatures of samples containing urea are shifted approx. 2 °C towards colder temperatures. This phenomenon is attributed to the freezing point depression induced by urea, which is also visible in the homogenous freezing regime. The sensitivity of the sample to urea treatment indicates proteins to play a role in the ice nucleation activity. Unfolding using urea as a reagent and further cutting peptide bonds decreases the INA. This suggests that the secondary and primary structure is important for the proteins to act as INMs.

However, in literature only Tong et al, (2015) deal with proteins as INMs in birch pollen. Still, the exact chemical composition of the INM from primary biological aerosol particles (PBAP) is a matter of controversial discussion. When Pummer et al., (2012) discovered that ice nuclei are water soluble and can be extracted from pollen, they proposed that polysaccharides are the responsible moieties. This was supported by infrared and raman spectra and by size filtration experiments pointing to INMs larger than 100 kDa bearing carboxylate groups (Pummer et al. 2013, Dreischmeier 2017). INMs with similar physical and chemical properties were also extracted from other PBAPs such as fungi (Haga et al. 2013, 2014, Kunert et al. 2019). This led Hiranuma et al. (2015, 2018) to the conclusion that cellulose, which is a polysaccharide and part of PBAP cell walls, might be

responsible for the INA. But also for other polymers such as lignin and nanogels, which can take up water in their structures as well, a similar ice nucleation activity as for cellulose was found (Bogler and Borduas-Dedekind 2020, Xi et al. 2021). Structure and size are also crucial for bacterial ice nucleation proteins, where repeat numbers (similar sequence patterns) and oligomerization contribute in a seemingly independent manner to the nucleation mechanism (Ling et al.2018, Šantl-Temkiv 2015). In general, INMs in the atmosphere are manifold and found from different sources such as forests, deserts and marine environments (Lloyd et al., 2020).

In summary, our experiments come to a similar conclusion as Tong et al. (2015) pointing in the direction of proteins but are not a priori in contradiction with the other recent literature. The missing piece of the puzzle could be a glycoprotein, which exhibits carboxylate functionalities, can bind water in tertiary structures and displays degeneration and unfolding of its secondary structure due to heat treatment or reaction with enzymes. Also glycoproteins are known to be part of the metabolism of cells regulating freezing stress tolerance (Zachariassen and Christensen, 2000). We suggest that further investigations should aim in this direction.

## 4. Conclusions

In this study we develop an extraction method that gives access to the cytoplasmic material of pollen grains, even after the grains have lost the ability to germinate and rupture. We emphasise that birch pollen contain soluble and insoluble cytoplasmic materials. The soluble INMs are easily extracted with water. In nature, INMs from the surface of birches are washed down by rain (Seifried et al., 2020), possibly ending up in soil and/or rivers (Borduas-Dedekind et al., 2019; Knackstedt et al., 2018). The insoluble material is mostly starch granules that we refer to as SPP in accordance with the usage of the term SPP in the medical sciences (e.g. Bacsi et al., 2006). INM are exclusively found within the aqueous solution but it took several times of washing and dilution for a sample to lose the ice nucleation activity suggesting that INM are present in high concentrations. In contrast, we find that purified SPP are not ice nucleation active. Fluorescence spectroscopy reveals a strong protein signal in the remaining ice active solution that is not found with the highly concentrated SPP. Generally, we see a quantitative link between the ice nucleation activity (CNC (-25°C)) and the protein concentration during the washing procedure. Therefore, the INM as well as the proteins of the suspension must be soluble and extractable during the filtration process in the same manner. We highlight that the ice nucleation activity of Betula pendula pollen is linked not only to polysaccharides (Pummer et al., 2015) but also to proteinaceous INM. The abundance of INM suggests that INM and SPP might not naturally separate in the atmosphere. SPP could act as carriers of INM. Similarly, starch granules are known to act as carriers of allergens of different pollen grains (Bacsi et al., 2006; Schäppi et al., 1999; Staff et al., 1999). Pollen rupture generating airborne SPP could therefore be a possible mechanism of how INM from pollen grains could disperse in the atmosphere without the presence of the grains.

### Author contribution

JB wrote the manuscript with contributions from all co-authors. JB and HG developed the concept of the study. JB and HG acquired financial support for the project. JB worked out the details and guided experiments. JB conducted pollen rupture experiments, took SEM images and produced the video material with the help of JG. JG and JB developed the extraction method. JG analysed subpollen particles and created the graphical representation. TS and PB conducted the ice nucleation measurements and respective data analysis including graphical representation. TS and PB performed the Bradford Assay, the fluorescence spectroscopy and the control experiment, as well as the digestion and unfolding experiments.

**Data availability:** All data are available from the corresponding author upon request

**Competing interest:** The authors declare that they have no conflict of interest.

**Acknowledgement**

The authors gratefully acknowledge the Hochschuljubiläumsstiftung der Stadt Wien (grant: H-318313/2018 and H-2075/ 2010) and the Austrian Science Fund (FWF; grant P 26040) for financial support. The authors would also like to thank Robin Hoheneder at the University of Natural Resources and Life Sciences, Vienna, for his help with the Bradford assay.

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

**Table 1. Protein concentration and cumulative nuclei concentration (CNC). LOQ = limit of quantification.**

| Sample | Protein Concentration [µg ml$^{-1}$] | CNC (-25°C) [pl$^{-1}$] | CNC (-34°C) [pl$^{-1}$] |
|--------|--------------------------------------|-------------------------|-------------------------|
| A | 77.4 | 7.6 | 13.2 |
| B | 25.8 | 4.5 | 13.1 |
| C01 | 5.3 | 4.0 | 8.7 |
| D | < 2.5 (LOQ) | 0 | 0 |

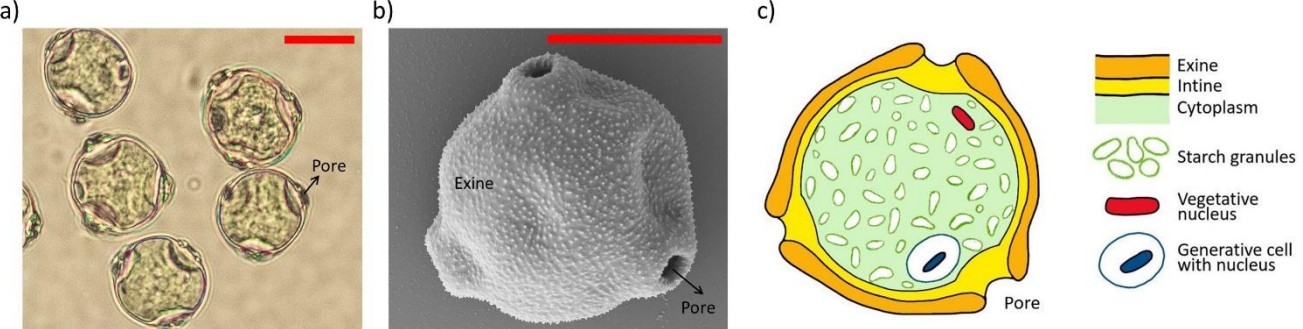

**Figure 1: Different views of birch pollen grains. a) Light microscope image of pollen grains in water. b) Electron microscope image of a single pollen grain. c) Schematic sectional view with the main components of a birch pollen grain. Red scale bar is 10 μm.**

**(a)**

Step 1:

Ultrapure Water

SPP

Step 2
→ Sample A

Paper filter

Solution with SPP (sample A)

Step 3
→ Sample B

Syringe filter

Solution without SPP (sample B)

Step 4
→ Sample C

Ultrapure Water

Water and Solution (sample C)

Step 5
→ Sample D

Ultrapure Water

Filter in reverse

SPP in Water (sample D)

**(b)** After Step 1

**(c)** After Step 5 (Sample D)

**Figure 2: a) Extraction process of SPP. Step 1: Entire pollen grains mixed with ultrapure water are crushed in a mixer mill (MM400, Retsch Gmbh, Haan, GER). Step 2: The sample is filtered so that entire pollen grains and large outer shells are excluded. Step 3: Filtration with a 0.2 μm pore sized filter. Step 4: The filter including the SPP is rinsed with ultrapure water to wash off soluble material. Step 5: The filter is reversed and the SPP are extracted with ultrapure water. b) Crushed pollen grains, cytoplasmic material and SPP (starch granules) after step 1. c) Extracted SPP after step 5. Red scale bar = 20 μm.**

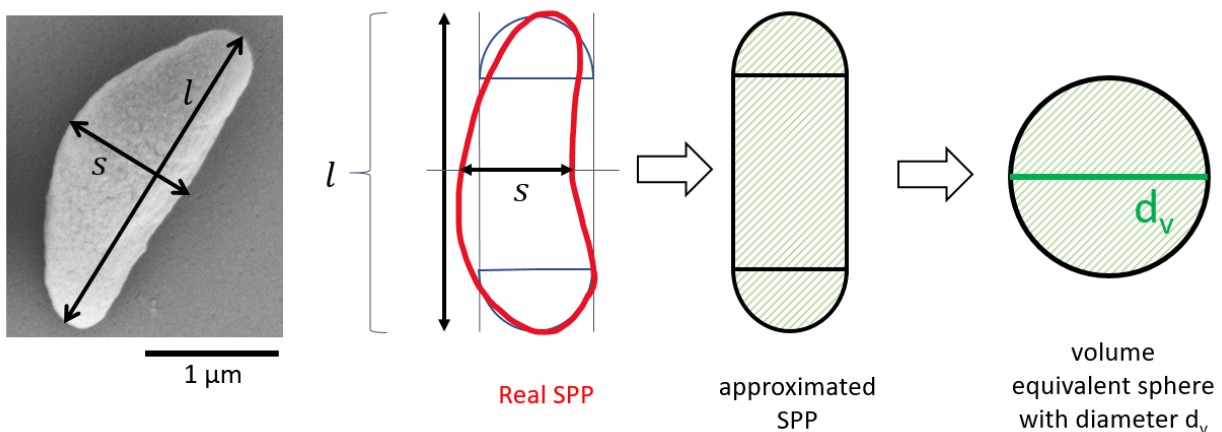

**Figure 3: Approximation of the shape of an SPP and determination of the respective volume equivalent diameter $d_v$.**

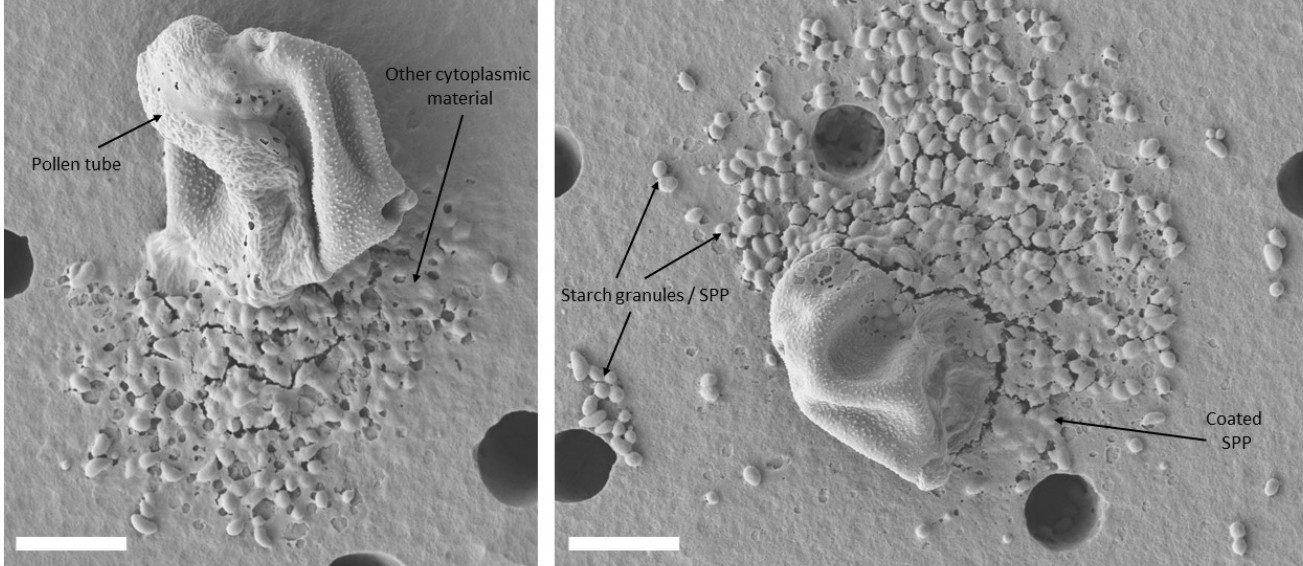

**Figure 4: Germinated and ruptured pollen grains. Freshly harvested birch pollen grains were exposed to high relative humidity. Pollen grains were deposited on a polycarbonate nuclepore filter (pore size: 8 µm) within an inline filter holder and humid air (95% relative humidity) was sucked through for 8 hours. This process can only be observed with**
725 **fresh pollen grains. Commercial pollen grains remained intact after the same treatment. White scale bar is 10 µm.**

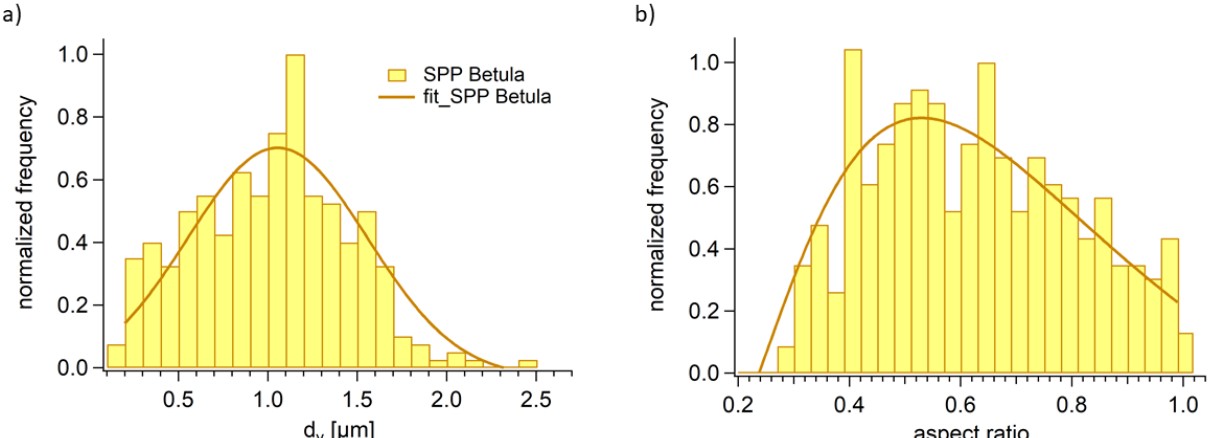

**Figure 5: a) Size distribution of SPP. The volume equivalent diameter dv was calculated by equation (1) that is based on the approximated shape of an SPP. The $d_v$ of 326 SPP was binned into 24 equidistant intervals between 0.2-2.5µm and normalized to the interval with the maximum number of SPP. b) Aspect ratio of SPP.**

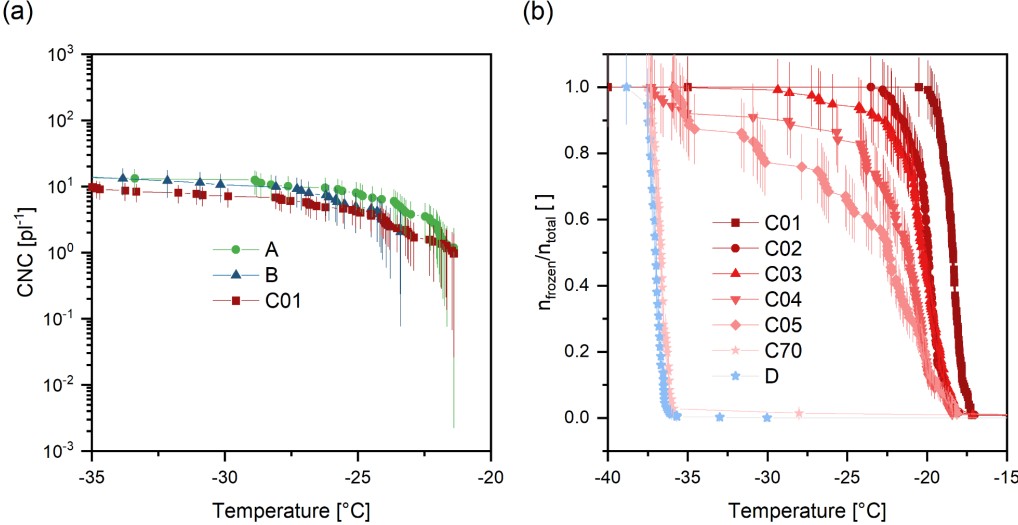

**Figure 6: Ice nucleation spectra of extracted samples. (a) Cumulative ice nucleus spectra of sample A, B and C01. Error bars correspond to the Gauß error propagation. (b) Freezing spectra of SPP washing solutions (C01, 1 ml water used to wash SPP, C02, second 1 ml fraction, etc.) and of the purified SPP (D). Vertical lines correspond to the counting**
**error. The temperature uncertainty of VODCA is 0.5 °C.**

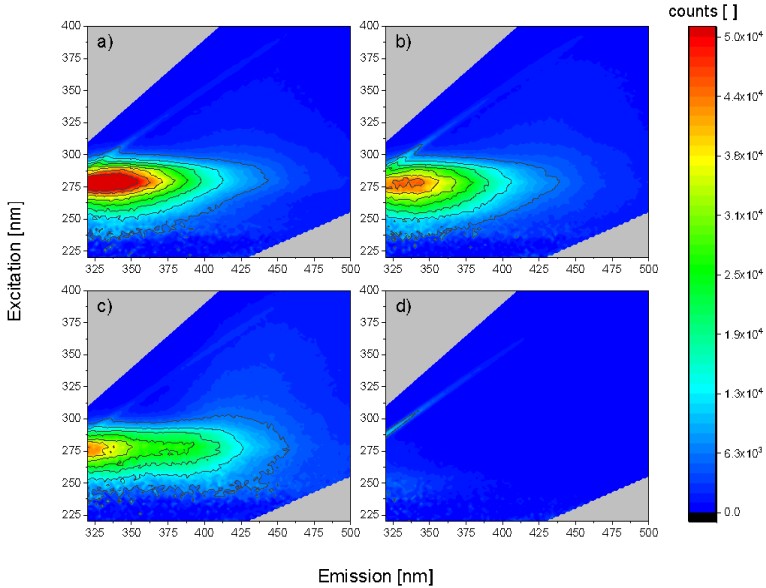

**Figure 7. Fluorescence excitation-emission maps of obtained SPP samples. a) Sample A, 1:10 diluted with ultrapure**
**water. b) Sample B, 1:10 diluted with ultrapure water. c) Sample C01. d) Sample D (washed SPP).**

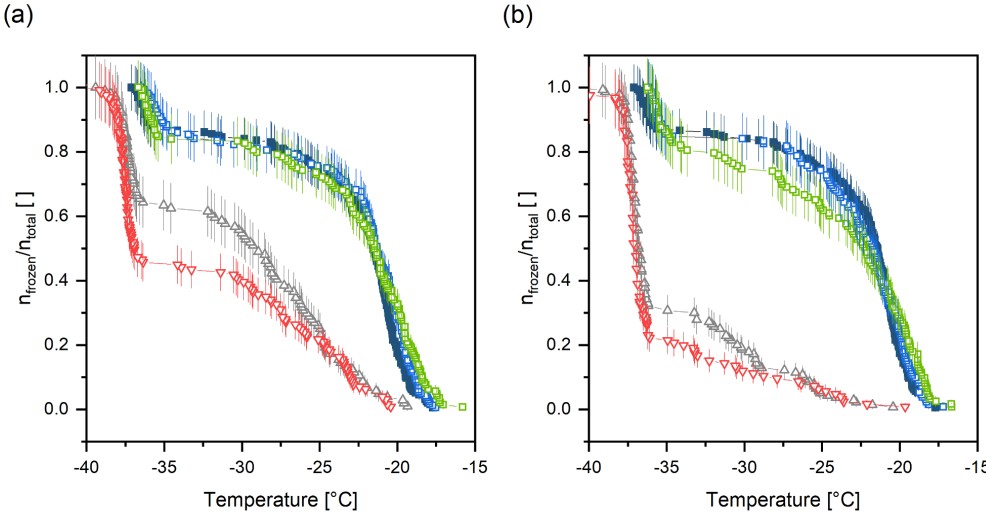

Figure 8. Results of the unfolding experiment and enzymatic digestion of extracted INMs (sample B) at 37 °C using subtilisin A (from *Bacillus licheniformis*) and urea after (a) 1 h and (b) 6 h incubation time. Filled, blue squares correspond to sample B in Tris buffer at room temperature prior to treatment and hollow, blue squares to the control sample. Green circles show the treatment with subtilisin A, grey triangles (cone up) with urea and red triangles (cone down) represent the treatment using a mixture of subtilisin A and urea. Vertical lines represent the counting error.
