# Peer review of "Isolation of subpollen particles (SPP) of birch: SPP are potential carriers of ice nucleating macromolecules"

_Biogeosciences, 2021_

## Author Comment (AC1)

We thank referee #1 very much for the extensive review of our manuscript and the positive feedback. We also much appreciate the critical feedback and the suggestion to add a broader discussion of related literature. Addressing comments and suggestions clearly improved the quality of our manuscript.

The comments and our answers are listed below. The referee's comments are written in black, our responses in green.

Overview of the work:

The authors present a three-part study where they first developed a method to extract SPP from pollen grains, then measured the ice nucleating ability of SSP and its washing water, and finally, quantified the protein content of the INMs from the washing water. The authors investigated the ice nucleating ability of different extraction steps of the pollen grain, and found that the SPP themselves were not ice active. However, the authors determined that the washing of the SPP resulted in the isolation of ice nucleating macromolecules. This procedure is well explained and well-illustrated and the investigated work is clearly described. Furthermore, the authors used two methods, fluorescence spectroscopy and quantitative protein analysis assay, to determine the protein content of the ice nucleating macromolecules. This last chemical step is of particular value to the research field of biological ice nuclei. The work presented merits publication in Biogeosciences after minor revisions; my suggested revisions are focused on additional experimental details as well as on adding important literature context and comparisons to the results and discussion sections.

General feedback:

1. Additional details on the replicates and on the uncertainty of the ice nucleation results are necessary. Can the authors comment on the number of replicates necessary to generate Figures 4. What are the uncertainties associated with the immersion freezing technique and can the authors add appropriate error bars to their freezing data in Figure 4?

   We thank the referee for this comment. Indeed, it is important to clarify the uncertainties of our ice nucleation assay. The temperature uncertainty of VODCA is 0.5 °C which was previously determined by Zolles et al., 2015. Furthermore, we determined the measurement uncertainty by calculating the counting error and performed a Gauß error propagation as described in Kunert et al., 2018. Errors larger than 100% were excluded in the graphs.

   **References:**

   Kunert, A. T., Lamneck, M., Helleis, F., Pöschl, U., Pöhlker, M. L., & Fröhlich-Nowoisky, J. (2018). Twin-plate Ice Nucleation Assay (TINA) with infrared detection for highthroughput droplet freezing experiments with biological ice nuclei in laboratory and field samples. Atmospheric Measurement Techniques, 11(11), 6327-6337.

Zolles, T., Burkart, J., Häusler, T., Pummer, B., Hitzenberger, R., & Grothe, H. (2015). Identification of ice nucleation active sites on feldspar dust particles. The Journal of Physical Chemistry A, 119(11), 2692-2700.

2. The work is well presented but lacks discussions on the comparison of the results with previously published work in the results sections and conclusion. For example, do the results presented corroborate (or not) the work by (Dreischmeier et al., 2017)? In the results section, can the authors elaborate on the comparison of their findings with work on fungal spores (Haga et al., 2014, 2013; Kunert et al., 2019)? Can the authors discuss their findings in terms of structural studies of INMs such as (Ling et al., 2018; Šantl-Temkiv et al., 2015)? The authors mention how the intine is composed of cellulose, yet there is interesting literature in ice nucleation on the relevance of cellulose which can be mentioned (see (Hiranuma et al., 2015, 2018)). Is there any connection with SPP and lignin (see (Bogler and Borduas-Dedekind, 2020))? In an atmospheric implication section, can the authors discuss if airborne ice nuclei of unknown origin could be attributed to SPPs (like in (Lloyd et al., 2020))? Are the INMs in this work forming nanogels (see (Xi et al., 2021))? In the opinion of the authors, is the field converging towards proteinaceous material is the most important INMs? If not, why? If so, why? Discussing these details will place the presented work in the greater context of the current literature on INMs and will allow future work to more effectively build upon the work presented.

We thank the referee for the feedback and agree that a deeper discussion regarding the chemical analysis of INMs will improve the manuscript. We extended chapter 3.3 Chemical Analysis not only with a deeper discussion but also with an additional experiment where proteins of sample B were unfolded using urea as a reagent and subtilisin as an enzymatic digestion tool (please see the details about the experiment in answer 1 to referee 2). Furthermore, we decided to remove the sub-headlines 3.3.1 and 3.3.2.

We added the following discussion to chapter 3.3 starting in line 268:

[revised manuscript text omitted]

Zachariassen, K. E., & Kristiansen, E. (2000). Ice nucleation and antinucleation in nature. Cryobiology, 41(4), 257-279.

Specific comments:

Title: I'm wondering if a title highlighting all three parts of the study (SPP extraction and INM isolation) may be more representative of the work. In addition, the major finding of the work is the chemical identification of proteinaceous material, which should also be mentioned in the title.

Would the authors consider revising their title along the lines of "Isolation of subpollen particles (SPP) and their ice nucleating ability: SPP are carriers of proteinaceous ice nucleating macromolecules"?

We thank the referee for the suggestion and decided to change the title. The new title now is: "Isolation of subpollen particles (SPP) of birch: SPP are potential carriers of ice nucleating macromolecules."

Abstract:

Are the authors interested in citing the thunderstorm asthma literature to *motivate* their research? As it currently stands, the mention of thunderstorm asthma appears to be an afterthought in the introduction. I would recommend mentioning this context already in the abstract as well as move the introduction paragraph (lines 87-99) earlier. It's an arguably important motivation for the presented research. Recent reference (including refs therein): (Bannister et al., 2021)

We decided to mention thunderstorm asthma only as an afterthought, as the phenomenon of thunderstorm asthma is predominantly related to grass pollen. To our knowledge no case of thunderstorm asthma in relation to birch pollen has been reported.

Line 13: the authors state a gap in knowledge, "explanations on how these materials could distribute in the atmosphere are missing" but do not address this gap in their study. Perhaps this sentence can be reworded to address the gap indeed addressed here.

We reworded the sentence accordingly: "However, INM and SPP are not clearly distinguished. This has motivated the present study which focuses on birch pollen and investigates the relationship between pollen grains, INM and SPP."

Line 16: what is meant by "loosely attached"? Van der Waals forces? Covalent bonds?

We could separate the INM from the SPP simply by rinsing the SPP with water. Due to the removability without reagents we believe that INMs are not covalently bonded. We decided to clarify the statement and reworded the sentence: "We show that INM are not bonded (i.e. can be washed off with water) to SPP."

Introduction:

Lines 69-70: I would be interested to read a (Tong et al., 2015) discussion in the results section in more details.

We agree with the referee and added a discussion about the chemical analysis of INMs including a discussion about proteins (Tong et al., 2015) in chapter 3.3. starting in line 268 (please see the text in the comment above; point 2).

Lines 105-109: I would suggest also placing the emphasis of this work on the identification of proteinaceous material as the ice active INMs. The authors show clear chemical evidence of these types of molecules, and this finding is interesting and important.

We agree with the referee that the finding of proteinaceous material as the ice active INMs is important. To prove the findings, we conducted an additional enzymatic digestion experiment. At the end of the introduction in line 109 we also added a sentence: "To clarify the role of proteins in heterogeneous freezing we conducted a specific enzymatic digestion/ protein unfolding experiment."

Methods:

Lines 116-118: add the date collection information.

We added the date of collection and the exact location of the birch tree. The sampling location has changed as we now refer to the pollen we used for the additional experiments (based on the suggestions of referee 2): "Freshly harvested pollen samples were collected from a birch tree at the Donaukanal (location birch tree: 48.237480, 16.362990) in Vienna. Collection took place on April 9, 2021."

Lines 145-150: I wondered whether a figure with the described shapes could be helpful for the reader to visualize this calculation.

We think this is a good idea and decided to add a figure to visualize the calculation. In the text we also added the equation used to calculate the volume equivalent diameter.

"Based on the approximated shape the volume equivalent diameter $d_v$ was calculated by equation:

$$d_v = s \sqrt[3]{\frac{1}{2}(\frac{3l}{s} - 1)} \qquad\qquad (1)"$$

[Figure]

**Figure 2: Approximation of the shape of an SPP and determination of the respective volume equivalent diameter $d_v$.**

Line 155: can the formation of the emulsion be described in more detail?

The experimental procedure was published already in 2012 by Pummer et al., 2015 by Zolles et al., and 2018 by Felgitsch et al. To highlight the linkage, we added in line 155: "[...] a detailed description of the experiment is given in Felgitsch et al., 2018, Zolles et al., 2015 and Pummer et al., 2012."

Furthermore, we described the preparation of the emulsion more clearly. We added in line 156 the mass percentage and the manufacturer/distributor: "(10 wt% lanolin, anhydrous, VWR Int., Radnor, PA, USA; 90 wt% paraffin, light, pure grade, AppliChem GmbH, Darmstadt GER)" and the information "by mixing with a pipette tip" to the sentence.

The improved description is stated as follows:

"INM content from birch pollen was quantified in immersion freezing mode by using the Vienna Optical Droplets Crystallization Analyzer (VODCA) setup (a detailed description of the experiment is given in Felgitsch et al., 2018, Zolles et al., 2015 and Pummer et al., 2012): An emulsion of 2 µl sample solution and 4 µl inert oil-mixture (10 wt% lanolin, anhydrous, VWR Int., Radnor, PA, USA; 90 wt% paraffin, light, pure grade, AppliChem GmbH, Darmstadt GER) is prepared by mixing with a pipette tip on a microscopic glass slide and transferred into a cryo-cell. Sample emulsions are cooled with a rate of 10 °C min$^{-1}$. The freezing process was monitored by videos at four different spots of each sample glass slide via a microscope camera (MDC320, Hengtech, GER)."


Line 262: What is the mass percentage of the sample that is thought to contain these concentrations of proteins?

Unfortunately, we were not able to determine the dry mass of our samples due to low concentration and limited sample volume.

Figure 1: it's not clear in the caption where images of 1a and 1b are from? Could additional details be added? Which part of the figure is copyrighted?

We realized that the phrase 'information for the drawing is taken from....' is confusing and overcautious. We deleted the phrase. All three images were created by the authors.

Figure 2: very clear! Well done! ï Small addition: could the instrument be specified in the caption as well?

We thank the referee for this feedback. We added the name and details of the instrument in the caption: "Extraction process of SPP. Step 1: Entire pollen grains mixed with ultrapure water are crushed in a mixer mill (MM400, Retsch Gmbh, Haan, GER). [...] "

Figure 3: additional details in the caption of how these distributions were generated would be useful.

We added details to the caption of Figure 3. The caption now reads:

Figure 3: a) Size distribution of SPP. The volume equivalent diameter $d_v$ was calculated by equation 1 that is based on the approximated shape of an SPP. The $d_v$ of 326 SPP was binned into 23 equidistant intervals between 0.2-2.5 µm. b) Aspect ratio of SPP.

Figure 4: Compelling data. Good job. Related to the general feedback, a discussion of replicates and uncertainties could be reported here in the caption and displayed on the graphs.

We thank the referee for these remarks. As mentioned above we added a statement regarding the uncertainty of the ice nucleation assay in chapter 2.4 and added error bars to the graphs.

Figure 5: Cool data!

We thank the referee for this positive feedback.

SI: the SI for this manuscript is quite short, and I wondered whether the authors might consider including the SI into the text (easier for the reader). For example, Figure S1 could be merged within Figure 2 of the main text.

We leave the control experiment in the SI but added the additional experiment to the main text.

Conclusion:

Line 274: can the authors make connections between the water-soluble component to dissolved organic matter? (see (Borduas-Dedekind et al., 2019; Knackstedt et al., 2018))

We thank the referee for this remark and added the following to line 273:

"The soluble INMs are easily extracted with water. In nature, INMs from the surface of birches are washed down by rain (Seifried et al., 2020) certainly ending up in soil and/or rivers (Borduas-Dedekind et al., 2019; Knackstedt et al., 2018)."

[revised manuscript text omitted]

---

## Author Comment (AC2)

We thank referee #2 very much for the detailed and critical feedback. Inspired by the comments and suggestions we have conducted additional measurements and rewrote sections of the manuscript to clarify statements. We believe that the quality of our manuscript has now greatly improved.

The comments and our answers are listed below. The referee's comments are written in black, our responses in green.

*This is an interesting manuscript describing experiments regarding the preparation and release of molecular solutes and small subpollen particles (SPP) from birch pollen. Some of the results, for example the characterization of the SPP are novel, well described and interesting, while some other experimental procedures and results are described only poorly. Moreover, I am questioning some of the interpretation regarding the connection between SPP and ice nucleating molecules (INM), and whether the INM are proteinaceous or not.*

*The data presented in the current manuscript might be quite useful and, hence, may support publication. I do have, however, major concerns regarding some of the procedures and the interpretation of the experiments and their application to the atmospheric situation, see below. Moreover, in some places the manuscript is technically deficient, e.g. in the detailed description of procedures or results, sometimes making it hard to understand what the authors actually refer to. Also, the citations and reference list need more care.*

*Overall, I think that major revisions are certainly required, as outlined below, before the paper may become suitable for publication in Biogeosciences.*

**Scientific points:**

*1.)  As far as I understand, cytoplasm generally contains also all kinds of soluble material such as proteins, lipids, polysaccharides, DNA, RNA and inorganic ions. Using the washing procedure in the extraction method described in Figure 2, I would assume that with more and more extraction volume in step 4, these solutes become more and more dilute in samples Cx. This is apparently the case for the INM. The authors also show a very slight decrease in CNC (meaning INM concentration) between samples A, B and C01. I am convinced one would see a very similar pattern with all other soluble materials, too, i.e. with proteins, with polysaccharides, with DNA, with ions etc. The authors of the current manuscript chose to investigate the content and concentration behavior of proteins using two methods (fluorescence and Coomassie staining), which is fine. And indeed, they found a similar concentration behavior for the proteins when comparing samples A, B, and C01. (They did not analyze whether proteins decrease in the same way as do INMs in samples Cx, though.) From the concentration correlation between INM and proteins in samples A, B, and C01, they infer that the INM may be proteins.*

*But what if the INMs belong to another substance class, for example DNA or polysaccharides? (The latter has been suggested in previous publications, also by the current authors.) I would assume to observe a similar correlation: The INMs are somewhat diluted between samples A, B*

*and C01. In that case, the correlation between proteins and INMs would be fortuitous and would just indicate a small dilution of (any kind of) soluble material from sample A, to B, and C01. Or another thought experiment: if the authors had chosen to measure the concentration of DNA rather than that of proteins, wouldn't they have observed a similar concentration trend for DNA, even if the INMs had been the proteins? That would not directly imply that the INMs are DNA, wouldn't it? So from my standpoint, the correlation between INM concentration and protein concentration is not a proof that the INMs are proteins. This fact should be stated more clearly in the conclusions and abstract, to avoid a false interpretation by readers. The sentence in lines 280-281: 'We highlight the possibility that the ice nucleation activity of Betula pendula pollen is linked not only to polysaccharides (Pummer et al., 2015) but also to proteinaceous INM.' goes in the right direction, but may not be enough.*

*We agree with the referee that we would possibly observe a similar trend for other material from the cytoplasm (e.g. DNA, polysaccharides). However, we see a strong single signal in the fluorescence measurements. That is the first indication that proteins are present in our extracts and samples. This is confirmed with the Bradford assay. Still, we agree with the referee that there is no proof for the INM to be from proteinaceous origin.*

*To clarify we conducted an additional experiment with sample B. In this experiment the proteins of the sample were unfolded using urea as a reagent and subtilisin as an enzymatic digestion tool. The full experimental details were added to the manuscript (see new section 2.7 below).*

*We found that the unfolding procedure with urea decreases the ice nucleation ability significantly. Furthermore, the strongest reduction is observed for a combination treatment with subtilisin and urea (unfolding and digestion of amide bonds). The results are added to the manuscript and clearly indicate that proteins play an important role in the observed freezing behavior of the sample solution. However, proteins are not thought to be the only INM in the chemically broad cytoplasm.*

*We thank the referee for his or her concern regarding the statement that proteins are INMs in birch pollen extracts. However, we believe that the additional experiment supports the hypothesis that the majority of INMs are of proteinaceous nature.*

*We added the following to the paper:*

*We inserted a subsection in the methodology explaining the experiments' details (line 193):*

*"**2.7 Enzymatic and chemical treatment***

[revised manuscript text omitted]

*Rawlings, N. D., Barrett, A. J., & Bateman, A. (2010). MEROPS: the peptidase database. Nucleic acids research, 38, D227-D233.*

*Kozloff, L. M., Turner, M. A., & Arellano, F. (1991). Formation of bacterial membrane ice-nucleating lipoglycoprotein complexes. Journal of bacteriology, 173(20), 6528-6536.*

*Pummer, B. G., Bauer, H., Bernardi, J., Bleicher, S., & Grothe, H. (2012). Suspendable macromolecules are responsible for ice nucleation activity of birch and conifer pollen. Atmospheric Chemistry and Physics, 12(5), 2541-2550.*

*Felgitsch, L., Bichler, M., Burkart, J., Fiala, B., Häusler, T., Hitzenberger, R., & Grothe, H. (2019). Heterogeneous Freezing of Liquid Suspensions Including Juices and Extracts from Berries and Leaves from Perennial Plants. Atmosphere, 10(1), 37.*

*Fröhlich-Nowoisky, J., Hill, T. C., Pummer, B. G., Yordanova, P., Franc, G. D., & Pöschl, U. (2015). Ice nucleation activity in the widespread soil fungus Mortierella alpina. Biogeosciences, 12(4), 1057-1071.*

*Gute, E., David, R. O., Kanji, Z. A., & Abbatt, J. P. (2020). Ice Nucleation Ability of Tree Pollen Altered by Atmospheric Processing. ACS Earth and Space Chemistry, 4(12), 2312-2319.*

*2.) Lines 206-209: 'However, even after 1-hour ultrasonic treatment we did not find any ruptured pollen grains nor SPP (Figure S1). We believe that the usually applied extraction method, where pollen grains are only left in water and are then filtrated, do not actually yield SPP unless very fresh pollen grains are used. In this sense our method is unique and offers the possibility to study isolated SPP and gain further insight about the location of the INM within the pollen grain.'*

*These sentences suggest that in all previous studies on dried commercial pollen, SPP were not present in the washing water. Is this notion correct? Please make a clear statement. If so, doesn't this imply that the INM are NOT connected to the SPP as in previous studies INM were indeed found by simply washing the dried 'old' pollen. Please discuss in more detail.*

*We agree that this notion is correct. We rewrote section 3.1. to clarify statements and added a more detailed discussion including additional literature.*

*The new section now reads:*

[revised manuscript text omitted]

3.) *Lines 253-254: 'The signal correlates with heterogeneous ice nucleation of sample A, B and C01'. I am not sure I entirely understand what is meant by 'correlates' in this context. Simply, samples A, B, and C01 show fluorescence and they also show ice nucleation? Or the ice nucleation activity, namely CNC or T_50, correlates with the strength of the fluorescence signal? Please explain in more detail.*

*We thank the referee for her/his feedback. CNC values of samples A, B and C01 follow the same trend as the fluorescence intensity at the maximum. We changed the wording in lines 253-254 for clarification to the following: "The fluorescence intensity at the maximum decreases with decreasing CNC (Table 1)."*

4.) *Lines 271-272: 'In this study we develop an extraction method that gives access to the cytoplasmic material of pollen grains, even after the grains have lost the ability to germinate and rupture.'*

*While I applaud the authors for the realization and description of this extraction method for dried pollen, I am missing an analysis / a connection of the results presented here to the processes occurring in the atmosphere. The authors make a big point that the release of cytoplasmic material in fresh ('living') pollen is different from that of dried ones. How can they then make any quantitative conclusions and statements on free pollen and their release of ice nucleating material?*

*We observe that fresh pollen grains directly released from catkins germinate and rupture when immersed in water or exposed to high relative humidity (>95 %). To illustrate the process, we now also made a video of freshly collected birch pollen grains immersed in ultrapure water. It can be seen clearly that particulate material is expelled from the pollen grains. Additionally, we exposed fresh pollen grains to relative humidities above 90 % for several hours and also find ruptured*

*pollen grains. Looking at the material released by fresh pollen grains with an electron microscope we find particulate and amorphous material. We have now added electron microscope pictures of ruptured pollen grains to the manuscript. We again highlight that such subpollen particles are not found with commercial pollen grains. With our extraction method we aim to mimic the natural rupture process, and to specifically investigate the ice nucleation behavior of these insoluble subpollen particles that are omitted by commonly used aqueous extraction methods done with commercially purchased pollen grains. The main purpose was to explore the role of the insoluble SPP in ice nucleation activity. Insoluble SPP are not gained with commonly used extraction methods.*

*A proper quantitative comparison (e.g. of protein content or amount of SPP) of fresh/living pollen with commercial pollen is far beyond the scope of this study. This would involve many more additional steps, such as the proper purification of fresh pollen grains (from spores, plant debris etc.), the exact weighing of samples in all steps during the extraction process, and also an analysis of the storing effects of fresh pollen grains in the lab. For example, we have noticed that fresh pollen grains are less likely to rupture when stored in the lab for just a couple of days.*

**Moreover, in the sample preparation part (line 116: 'Freshly harvested pollen samples were collected from birch trees at the Danube Island in Vienna.'), the authors mentioned that they also collected fresh pollen in Vienna, but I did not see any comparative analysis or measurements of fresh with dried pollen. Why is that so? The authors could have made experiments with fresh pollen using the same SPP extraction procedure, using only steps 2-4, and then analyzing the filtrates in a similar manner. Why did they not do so?**

*We thank the referee for this comment. The goal of this work was firstly to define SPP, as there are many different definitions used in the literature. Further, we wanted to test the hypothesis whether SPP from birch pollen are ice nucleation active or not. We have observed that the behavior of fresh and commercial pollen in water and at high relative humidity differ. Fresh pollen rupture, germinate and release SPP, while commercial ones do not. We have now added a video and electron microscope images to better illustrate the behavior of fresh pollen. The lack of SPP with commercial pollen (when the usual extraction method is applied) is the most obvious difference between fresh and commercial pollen. However, fresh pollen is only available during a very short period of the year (roughly 2-6 weeks during the pollination season at a specific location). That is the reason why the scientific community uses mostly commercial pollen. In our study we wanted to draw comparisons to other studies and use a standardized sample (Betula pendula, Allergon, Sweden). We therefore developed an extraction method of SPP from commercial pollen and further focused our analysis on commercial pollen.*

**5.) Lines 281-285: 'INM and SPP are both contained in the cytoplasm. The abundance of INM suggests that INM and SPP might not naturally separate in the atmosphere. SPP could act as carriers of INM'**

**I was wondering whether the authors can really exclude that the observed INMs come from the outer part of the pollen. I again emphasize the fact that dried pollen release INM (as shown in previous publications), but not SPP (according to the authors' statements) contradicts the**

statement that INM and SPP are both contained in the cytoplasm. If INM come from the cytoplasm AND are released even without rupture, do we need to consider two different types of INM then? Please elaborate.

*We agree with the referee that we cannot entirely exclude that INMs come from the outer part of the pollen. At this point it is therefore not justified to distinguish between two different types of INM. We deleted the respective sentence in the conclusions.*

6.) I still have not understood whether the amount of washing water given in Figure 4 (and Figure S2 in the supplement) refer to cumulative volume values or not. For example, for sample C01, 1 mL of washing water was used, and hence sample C01 has a total volume of 1 mL. What about sample C02? Was another 1 mL of washing water used (cumulatively the second mL) and the total volume is again 1 mL? Or were 2 mL of water used for sample C02, giving a total volume of 2 mL, and making it cumulatively 2-3 mL of waters used. Similarly, is sample C70 10 mL in total volume with the cumulative 60-70 mL of washing water used (there is a sample C60 given in the supplement)? Please explain more clearly.

Along the same lines, I am not sure how the dilution factor in equation 1 was applied to the different sample Cx solutions, and also to the samples A, B, and D. If you use different water volumes for extraction/preparing samples A, B, Cx, and D, shouldn't the CNC concentration be quite different? Or was that volume taken into account in the dilution factor? If yes, which solution is the reference? Solution C01?

*After step 3 in the extraction process (see Figure 2a), the sample (retentate in the filter) was washed with 1 mL ultrapure water; the obtained filtrate was named C01. Next, the supernatant was washed again with 1 mL ultrapure water; the obtained filtrate was collected in a separate vessel and named C02. The number index of the samples Cx refers to the cumulative amount of ultrapure water used in the sequential washing procedure.*

*To clarify we changed the text in line 139: "The ice nucleation activity of each sample C fraction was tested for each rinsing step (note that the rinsing water was not pooled)." Furthermore, we changed the caption in Figure 2 to "(b) Freezing spectra of SPP washing solutions (C01, 1 mL water used to wash SPP, C02, second 1 mL fraction, etc.) [...]"*

*In general, all samples where we calculated the CNC value were diluted with ultrapure water prior to ice nucleation measurements to prevent an underestimation of INM concentration freezing at lower temperatures. Thus, samples which did not freeze partly homogeneously in the first measurement were diluted and re-measured (see Felgitsch et al., 2018). The results of CNC (diluted) measurements are shown in Figure 2a only. They should give an impression that the CNC value is not strongly influenced by filtration, i.e. the INMs pass every filter used in this study.*

*"The freezing process was monitored by videos at four different spots of each sample glass slide via a microscope camera (MDC320, Hengtech, GER). On each spot about 20 droplets in the corresponding size range are observed."*

12.) Lines 233-234 and Table 1: 'only after 70 mL of washing the ice nucleation activity is entirely lost' There are some data points at temperatures higher than -34°C, both in Figure 4 and Figure S2. Were they ignored in this statement? Also, in Table 1, the CNC values for sample D at -25°C and -34°C are given as zero. Again, I am surprised, because the n_frozen/n_total ratio in Figure 4 and Figure S2 shows values slightly larger than 0. Please elaborate.

*We thank the referee for this comment. We confirm that there are data points slightly above the background. Therefore, we elaborated the statement and change it to "After sample C10 the ice nucleation activity rapidly diminishes but only after 70 mL of washing homogeneous freezing was the dominant process in the experiment (99 % of the observed droplets froze homogeneously)"*

13.) Figure 1: It is not clear to me whether the images shown in panels a ) and b) and the sketch in panel c) are original to the current work, or whether they have been taken from the given references. I do not understand what is meant in the caption by 'information for the drawing is taken from....'

*The images and the sketch are original to the current work. We intended to point out that knowledge about the composition of a pollen grain was taken from the respective literature and is not original to the current work. This might be overcautious and to avoid confusion we deleted the notion "information for the drawing is taken from […]."*

Referee 1 stated: "Furthermore, the authors used two methods, fluorescence spectroscopy and quantitative protein analysis assay, to determine the protein content of the ice nucleating macromolecules."

I do not agree with the phrase "the protein content of the ice nucleating macromolecules". As far as I can see, the authors have shown that the soluble material released from the cytoplasm contains proteins, and quantified them, and that the cytoplasm also contains ice nucleating molecules. But they did not show that the proteins are the ice nucleating molecules. There is some concentration correlation between the proteins and the ice nucleating molecules, but I would argue that the same correlation would hold for any soluble molecules contained in the cytoplasm,

*also those that were not analyzed regarding their chemical nature (e.g., polysaccharides, DNA etc.).*

*We agree with the statement of the referee. In order to make a clear connection to proteins we have conducted additional measurements as described above. These measurements strongly indicate that the majority of INMs is of proteinaceous origin. However, we cannot fully exclude that other substances also contribute to the ice nucleation activity of birch pollen extracts.*

---

## Author Response (AR2)

Responses to referee #1

We thank Referee #1 for acknowledging the improvement of our manuscript.

The comments and our answers are listed below. The referee's comments are written in black, our responses in green.

The authors did a good job addressing the reviewer comments. I have small additional suggestions at this stage:

1. The discussion in added to chapter 3.3 starting in line 268 is well done. I would also encourage the authors to state their key conclusions: "The piece of puzzle could be a glycoprotein, which exhibits carboxylate functionalities, is larger 100kDa, can bind water in tertiary structures and displays degeneration and unfolding of its secondary structure due to heat treatment or reaction with enzymes." could also be added to the abstract.

We thank the referee for this suggestion and added the sentence to the abstract in line 27.

"The missing piece of the puzzle could be a glycoprotein, which exhibits carboxylate functionalities, can bind water in tertiary structures and displays degeneration and unfolding of its secondary structure due to heat treatment or reaction with enzymes."

2. Perhaps I can also further probe the authors on the idea of glycoproteins. Would they also explain the activity seen by dissolved organic matter in these papers (Borduas-Dedekind et al., 2019; Knackstedt et al., 2018; Moffett et al., 2018; Moffett, 2016)?

Theoretically, proteins might play an important role in the ice nucleation activity of dissolved organic matter. However, we have no specific information if this was the case in the here mentioned studies. We would propose to investigate those samples using advanced analytical methods such as fluorescence spectroscopy or mass spectrometry in future studies.

3. On the point of the comment on Line 16: what is meant by "loosely attached"? Van der Waals forces? Covalent bonds? The authors comment: "We could separate the INM from the SPP simply by rinsing the SPP with water. Due to the removability without reagents we believe that INMs are not covalently bonded. We decided to clarify the statement and reworded the sentence: "We show that INM are not bonded (i.e. can be washed off with water) to SPP." However I would argue that "washing with water" could also cause hydrolysis of ester bonds and therefore is not proof of the absence of covalently-bonded moieties. pH-dependent water washing could help shed light on this mechanism.

This is a good point. We agree that covalent bonds cannot be excluded entirely, since hydrolysis might have cleaved bonds during the rinsing process. We now stated clearly in line 301 that a definite statement cannot be made:

"However, based on our experiments we cannot make a definite statement on the type of bonding."

Borduas-Dedekind, N., Ossola, R., David, R. O., Boynton, L. S., Weichlinger, V., Kanji, Z. A., and McNeill, K.: Photomineralization mechanism changes the ability of dissolved organic matter to activate cloud droplets and to nucleate ice crystals, Atmospheric Chem. Phys., 19, 12397–12412, https://doi.org/10.5194/acp-19-

12397-2019, 2019.

Knackstedt, K., Moffett, B. F., Hartmann, S., Wex, H., Hill, T. C. J., Glasgo, E., Reitz, L., Augustin-Bauditz, S., Beall, B., Bullerjahn, G. S., Fröhlich-Nowoisky, J., Grawe, S., Lubitz, J., Stratmann, F., and McKay, R. M.: A terrestrial origin for abundant riverine nanoscale ice-nucleating particles, Environ. Sci. Technol., https://doi.org/10.1021/acs.est.8b03881, 2018.

Moffett, B., Hill, T., DeMott, P., Moffett, B. F., Hill, T. C. J., and DeMott, P. J.: Abundance of biological ice nucleating particles in the Mississippi and its major tributaries, Atmosphere, 9, 307, https://doi.org/10.3390/atmos9080307, 2018.

Moffett, B. F.: Fresh water ice nuclei, Fundam. Appl. Limnol., 188, 19–23, https://doi.org/10.1127/fal/2016/0851, 2016.

Responses to referee #2

We thank referee #2 for his/her additional comments.

The comments and our answers are listed below. The referee's comments are written in black, our responses in green.

(1) In response to my comment 4.) the authors wrote the following text: "We observe that fresh pollen grains directly released from catkins germinate and rupture when immersed in water or exposed to high relative humidity (>95 %). To illustrate the process, we now also made a video of freshly collected birch pollen grains immersed in ultrapure water. It can be seen clearly that particulate material is expelled from the pollen grains. Additionally, we exposed fresh pollen grains to relative humidities above 90 % for several hours and also find ruptured pollen grains." I cannot find this information on observations at high humidity, neither in the text nor in the supplement. Apparently, the authors have provided the video for review purposes only, but I recommend that it is made available to the readers directly as a supplement/asset.

Actually, we added the information in section 3.1, where we also provide a link to the video. We intend to provide the video also to the readers.

(2) Line 132/133: "With commercially purchased pollen this ability is almost lost." Can you provide any reasonable explanation for why this is the case? What is the difference to fresh pollen material?

Pollen rupture and germination are closely related to the viability of pollen grains. Viability can be lost due to desiccation (e.g. during storing conditions or by exposure to sunlight) and aging of the pollen grain (e.g. Stanley and Linskens 1974; Siriwattanakul et al. 2019).

We added a brief explanation and changed the sentence in line 134 to:

"With commercially purchased pollen this ability is almost lost due to aging and desiccation of pollen and the loss of viability."

Siriwattanakul, U., Piboonpocanun, S., & Songnuan, W. (2019). Rapid pollen rupture and release of pollen cytoplasmic granules upon hydration of allergenic grass and weed species commonly found in subtropical regions. *Aerobiologia*, *35*(4), 719-730.

Stanley, R. G., Linskens, H. F. (1974). *Pollen: biology biochemistry management*. Berlin, Heidelberg (Germany): Springer Science & Business Media

(3) Figure 2a: Please indicate in Figure 2a which part is defined as sample B, C etc. Currently, it is mentioned that Step 3 yields sample B and Step4 yields sample C, although in the first case the sample is in the filter and in the other case it is the filtrate. Please indicate this directly in each figure panel so that it is directly obvious to the reader without having to read the detailed text in lines 141-151.

We realized that the reference to sample B in the text is confusing. Actually all samples refer to the filtrate. We have now explicitly described sample B in the text and also clarified the labelling in figure 2a.

(4) Line 181: "Note that one ice nuclei can also be…" Use singular for nuclei, i.e. nucleus.

Done.

(5) Line 189: "Errors larger than 100% were excluded in the graphs." I do not understand this procedure, why is it done?

Indeed, the sentence is misleading. We excluded **data points** (not errors) with errors larger than 100% in the graphs. This approach is described in Kunert et al. (2018). We changed the sentence in line 189 to:

 "Furthermore, we calculated the counting error and performed a Gauß error propagation as described in Kunert et al. (2018). Accordingly, data points with errors larger than 100% were excluded in the graphs."

Kunert, A. T., Lamneck, M., Helleis, F., Pöschl, U., Pöhlker, M. L., & Fröhlich-Nowoisky, J. (2018). Twin-plate Ice Nucleation Assay (TINA) with infrared detection for high-throughput droplet freezing experiments with biological ice nuclei in laboratory and field samples. Atmospheric Measurement Techniques, 11(11), 6327-6337.